# Nanoparticle Classification Using Frequency Domain Analysis on Resource-Limited Platforms [note 1]

**DOI:** 10.3390/s19194138

**Published:** 2019-09-24

**Authors:** Mikail Yayla, Anas Toma, Kuan-Hsun Chen, Jan Eric Lenssen, Victoria Shpacovitch, Roland Hergenröder, Frank Weichert, Jian-Jia Chen

**Affiliations:** 1Department of Computer Science, TU Dortmund University, Otto-Hahn-Str. 16, 44227 Dortmund, Germany; anas.toma@tu-dortmund.de (A.T.); kuan-hsun.chen@tu-dortmund.de (K.-H.C.); janeric.lenssen@tu-dortmund.de (J.E.L.); frank.weichert@tu-dortmund.de (F.W.); jian-jia.chen@tu-dortmund.de (J.-J.C.); 2Computer Engineering Department, An-Najah National University, Nablus P.O. Box 7, Palestine; 3Biomedical Research Department, Leibniz Institute for Analytical Sciences, ISAS e.V., Bunsen-Kirchhoff-Straße 11, 44139 Dortmund, Germany; victoria.shpacovitch@isas.de (V.S.); roland.hergenroeder@isas.de (R.H.)

**Keywords:** nanoparticles, frequency domain analysis, mobile sensors, PAMONO biosensor, surface plasmon resonance, embedded systems

## Abstract

A mobile system that can detect viruses in real time is urgently needed, due to the combination of virus emergence and evolution with increasing global travel and transport. A biosensor called PAMONO (for Plasmon Assisted Microscopy of Nano-sized Objects) represents a viable technology for mobile real-time detection of viruses and virus-like particles. It could be used for fast and reliable diagnoses in hospitals, airports, the open air, or other settings. For analysis of the images provided by the sensor, state-of-the-art methods based on convolutional neural networks (CNNs) can achieve high accuracy. However, such computationally intensive methods may not be suitable on most mobile systems. In this work, we propose nanoparticle classification approaches based on frequency domain analysis, which are less resource-intensive. We observe that on average the classification takes 29 μs per image for the Fourier features and 17 μs for the Haar wavelet features. Although the CNN-based method scores 1–2.5 percentage points higher in classification accuracy, it takes 3370 μs per image on the same platform. With these results, we identify and explore the trade-off between resource efficiency and classification performance for nanoparticle classification of images provided by the PAMONO sensor.

## 1. Introduction

Due to the evolution and emergence of viruses, together with increasing global travel and transport, there is the risk of spreading epidemic diseases. Therefore, an accessible mobile real-time virus detection device is needed. One portable sensor for the detection of viruses in liquid samples is the Plasmon Assisted Microscopy of Nano-size Objects (PAMONO) biosensor [1,2]. It addresses many application scenarios, since it enables fast and quick diagnoses in settings such as hospitals, airports, and in the open air. The sensor provides a sequence of images, which are analyzed by an image analysis pipeline to detect the presence of viruses and virus-like particles (VLPs). Figure 1 shows image patches of positive and negative samples, which are cut out from full sensor images. The blob-like excitations in Figure 1a are induced by virus particle-to-antibody bindings, and the excitations in Figure 1b represent negative samples.

Lenssen et al. [3] proposed state-of-the-art solutions based on convolutional neural networks (CNNs), which have good classification and execution time performance on general purpose hardware. However, to be evaluated, CNNs require considerable resources, such as high-performance GPUs, due to their complex architectures [4,5]. Although research on developing dedicated hardware for the evaluation of the CNNs is in progress [5,6], there is still no general purpose solution with low power consumption available for mobile and embedded systems, to the best of our knowledge. Therefore, we evaluate modifications of the existing image analysis pipeline, which aim to drastically reduce the demand for resources while keeping detection performance high.

In this work, we modify the classification stage of the PAMONO image analysis pipeline presented by Siedhoff et al. [1] by employing a frequency domain analysis-based classification, which has been widely used in the literature to detect abnormalities in medical images [7,8]. Frequency domain analysis methods have been used for many decades, and efficient algorithms and hardware are well established. As examples of frequency domain analysis methods, we evaluate the use of Fourier and Haar wavelet features, i.e., fast Fourier transform (FFT) and fast wavelet transform (FWT) features. In contrast to CNNs, which have high resource demands, frequency domain analysis methods could enable the nanoparticle classification with high accuracy on mobile and embedded systems with strict resource limitations.

In our previous work in [9], we use a GPU to accelerate parts of our frequency domain analysis based classification, and compared it to the CNN in terms of execution time and classification performance using the same evaluation setting. The objective of the work at hand is to further decrease the resource demand of nanoparticle classification methods for the PAMONO biosensor, to make them more suitable for use in devices with strict resource constraints. Therefore, in this work, we report on extensive experiments exploring strategies to further reduce the resource demand of the nanoparticle classification system for the PAMONO biosensor. Our contributions can be separated into two parts. First, we explore the extraction of frequency-domain based features (and classify them with a small decision tree), and we employ feature analysis on the extracted features to find potential subsets of features such that there is negligible loss in classification performance. It may require fewer resources to extract the features during the run time of the system. Second, we explore several different platform configurations for the execution of the frequency-domain analysis based nanoparticle classification and compare them to each other, i.e., using: (1) a mobile Intel CPU with an integrated GPU as an accelerator; (2) a mobile Intel CPU-only version; (3) an ARM CPU-only version to evaluate on an embedded platform; and (4) an ARM CPU with programmable logic (PL) on a field-programmable gate array (FPGA) to accelerate the frequency domain feature extraction.

The results show that, using an Intel Core i7 4600U CPU-only version (without acceleration) for the frequency analysis based classification, it takes 29 μs per image for the Fourier features, and 17 μs for the Haar wavelet features. On an ARM Cortex A9 processor, the execution times are 276 and 88 μs, respectively. The classification performance is above 96.5% for all cases. For comparison, the CNN-based method uses a GPU as an accelerator and takes 3370 μs per image on an Intel Core i7 4600U using the integrated GPU, but scores 1–2.5 percentage points. higher in classification accuracy. The results show that the classification of virus particles can be run on platforms with strict resource requirements. We also identify the trade-off between execution time and classification performance. With the frequency domain based classification, the setup for feature extraction can be configured for either faster execution and less accuracy, or slower execution and more accuracy.

## 2. Materials and Methods

### 2.1. PAMONO Sensor

The PAMONO sensor is an optical biosensor that is used to detect nanoparticles, such as viruses or microvesicles, in liquid samples [2,10,11]. Nanoparticles are too small to be detected using light directly. The sensor, however, utilizes Kretschmann’s scheme [12] of plasmon excitation to detect the individual bindings of virus or virus-like particles to antibodies applied to a gold surface. A sketch of the setup and operation of the PAMONO sensor is shown in Figure 2. First, the sample is pumped with a rotatory pump or a syringe into the flow cell with the gold plate, onto which the viruses or virus-like particles (VLPs) can attach due to the antibodies on it. A laser is pointed to the gold surface such that a charge-coupled device (CCD) camera can record the surface over time. The camera produces image sequences that can then be analyzed. When a nanoparticle binding occurs on the gold surface, intensity steps can be observed in the image sequences. The intensity steps emerge on the sensor images as blob-like excitations with wave-like excitations around them. Figure 1a shows some example image patches.

### 2.2. PAMONO Image Analysis Pipeline

Figure 3 shows the modified image analysis pipeline. It can be separated into four main steps: preprocessing, particle detection, feature extraction, and classification. The core area of this work is the feature extraction and the classification stages, which are shown in the diagram by the dashed rectangle. The feature extraction and classification parts of the pipeline are considered as the frequency analysis-based nanoparticle classification. In this work, we aim to modify the image analysis pipeline starting from the end point, the feature extraction and classification stage, such that more image patches can be classified than the state of the art. If a more efficient method with similar classification performance is found, then the detection stage can be simplified as well, since it will be possible to classify a lot more image patches in the same amount of time.

The raw input image can vary in size depending on the given dataset. The dataset can be accessed on the project’s website in [13]. Possible sizes are, e.g., 1080×145, 706×167, and 450×170 pixels. The constant background noise signal is removed from the raw sensor image in the preprocessing stage, and the particle signals are extracted using the signal model proposed in [1]. In the detection stage, the processed image is divided into small patches of size 48×48. Different methods for the detection stage and their performance in generating patches are presented in [3]. The current state-of-the-art method for the detection stage is a fully convolutional neural network that uses a stack of 8 consecutive images from the data stream as input [3]. It detects the nanoparticle binding excitations by performing binary segmentation, outputting a two-class confidence map to which a marching squares algorithm is applied to produce image patches.

After the generation of 48×48 pixel patches, for the first step of the frequency domain analysis-based classification, 32×32 pixels images are sub-sampled from the center of the generated patches. They are considered as the input for our classification system. The images may contain blob-like excitations from bindings in the positive case, or signals, artifacts, and noise unrelated to binding excitations in the negative case. For the feature extraction stage, the images are provided to the classification system, which extracts features using frequency domain analysis. For the feature classification, we evaluate decision trees (DT) and random forests (RF) to decide whether excitations due to a particle binding were observed. DTs and RFs with limited depths are especially well suited for use in resource-limited platforms.

### 2.3. Platforms and Development Environments

We evaluated our frequency domain analysis-based classification on three different platforms. For the first setting, we used the Intel Core i7 4600U with 8 GB of RAM and offloaded Open Computing Language (OpenCL) kernels to its integrated Intel HD Graphics 4400 using the framework deepRacin [14]. The deepRacin framework is also used in [3] to accelerate CNNs computations to GPUs. To compare the performance of the feature-based method to CNNs, we implemented our classification pipeline as deepRacin computation graphs as well. We also implemented our classification pipeline as a CPU-only version on the Intel Core i7 4600U with 8 GB of random-access memory (RAM), the second platform, and on the ARM Cortex-A9 on the Zedboard, the third platform, to compare the execution time to the deepRacin implementation. The Zedboard was used as an embedded evaluation platform, which has flexible acceleration capabilities.

#### 2.3.1. deepRacin

The framework deepRacin was used for the first part of our experiments. In deepRacin, OpenCL kernels are used to offload computations to accelerators, such as GPUs. Computation graphs with tensor operations, i.e., neural network models or image processing pipelines, can be executed in deepRacin, e.g., to offload parallel computations for efficiency. The CNN computations were accelerated with deepRacin [3]. We implemented the fast Fourier transform FFT and fast wavelet transform FWT computations used in our feature extraction as OpenCL kernels, and we deployed them on the Intel HD Graphics 4400 GPU using deepRacin.

#### 2.3.2. Zedboard with Xilinx Tools

Since our image processing algorithms are tailored for small embedded systems, we also executed the classification on an ARM Cortex-A9 to evaluate whether the classification can be run efficiently on small low-power embedded chips. We chose the Zedboard for additional evaluations. The Zedboard is a heterogeneous Zynq SoC and multi-processor system-on-chip (MPSoC) platform, with the software-defined system-on-chip (SDSoC) development environment [15] for building our proposed image analysis pipeline. The Zedboard has an ARM Cortex-A9 processor and PL (Programmable Logic), a component that can be used to create reconfigurable digital circuits. The PL can be used for the acceleration of parts of the image processing pipeline.

SDSoC offers the xfft IP (Intellectual Property) core, which can be instantiated in the PL. The xfft IP can be executed with a C++ function call to accelerate the FFT. We evaluated the use of the xfft IP core v9.0 against a statically linked FFTW 3.3.8 library [16] in our image processing pipeline on the Zedboard.

SDSoC also provides automated software acceleration in PL, which means that functions in C++ can be accelerated by creating reconfigurable digital circuits for them, but we do not explore this further in this work.

### 2.4. Feature Extraction and Analysis

The 48×48 image patches that are provided by the detection stage are subsampled from the center as 32×32 patches and are considered as the input for the classification system. The image patches may contain particles in addition to the existing artifacts and non-constant noise. In the feature extraction stage, the images are loaded into the classification pipeline, which extracts features in the frequency domain. For classification of the features, decision trees and random forests were evaluated for deciding whether excitations due to a particle binding were observed.

Frequency domain analysis is a classical method for signal processing and has many application cases in the field of medicine [7]. The provided data from the PAMONO sensor are signals that show light intensity steps over time. In the previous work and in this work, the data were analyzed with methods commonly applied in image processing. The goal of this analysis is to detect blob-like and periodic patterns in the images from the PAMONO sensor. In Figure 1, positive and negative examples of input images can be seen. Images with particle bindings show blob-like excitations with surrounding periodic patterns resembling sinusoidal waves. Therefore, frequency analysis methods are proposed to classify particle bindings. As examples of frequency analysis techniques, the Fourier and Haar wavelet transform are used to extract features.

For the feature extraction, the input image is first transformed from the spatial domain into the target domain. Then, several features are extracted from the transformed and further processed image to extract texture information, using the information of which the type of pattern or excitation can be identified. A general overview of the feature extraction processes is presented in Figure 4, with the extraction of Fourier features in the upper branch and the extraction of wavelet features in the lower branch.

#### 2.4.1. Fourier or Spectral Features

In this work, the Fourier features are used to differentiate between periodic and non-periodic patterns in the image patches. With these features, it is possible to quantify the differences between the patterns [17]. Theoretically, every signal can be represented as a superposition of sinusoidal signals with different frequencies. The Fourier transform divides a signal into its frequency components, from which the original representation can be derived again. The two-dimensional Fourier transform, when integrated over $R^{2}$, produces the frequency representation of f(p):(1)$$F\left(w\right)=\int f\left(p\right)\;e^{-2\pi ipw}\;dp.$$

In this work, the two-dimensional discrete Fourier transform (DFT) is considered, since the inputs are grayscale images. Every pixel of the 2D transform represents a sinusoidal function with information about the phase, magnitude, and frequency. With an image *I* of dimension N×N and its Fourier transform *F* at (l,k), the DFT can be obtained by:(2)$$F\left(l,k\right)=\sum_{p_{x}=0}^{N-1}\sum_{p_{y}=0}^{N-1}I\left(p_{x},p_{y}\right)e^{-2i\pi (\frac{kp_{x}}{N}+\frac{lp_{y}}{N})}.$$

Instead of computing Equation (Equation 2) directly, the Fourier transform can first be applied on all the rows of the image, and then on all the columns of the image, because of the algorithm’s separability property. For the single rows and columns, a divide-and-conquer method, the fast Fourier Transform (FFT) algorithm, can be used for efficient computation.

For the feature extraction, the FFT algorithm is first performed on the input image. To compute spectral features from the Fourier transform of the input image, the zero-frequency component is shifted to the center, and magnitude of the spectrum is computed. Then two sets of magnitude values are extracted, which are referred to as $S_{rad}$ and $S_{ang}$. A visualization of the extraction of the Fourier features is shown in the upper branch in Figure 4. $S_{rad}$ is an array of size *r* and stores the sums of magnitudes over semicircles for different radiuses *r*. This means that $S_{rad}$ contains one entry for each semicircle with radius *r* that can be drawn into the image. $S_{ang}$ consists of sums over magnitudes lying on straight lines between the image center and the outer semicircle. $S_{ang}$ contains one entry for each straight line between the center and the outer semicircle. This way of summing is illustrated in Figure 4 by the blue and red lines, for $S_{rad}$ and $S_{ang}$, respectively. Positive and negative images can be distinguished by observing high-magnitude bursts in semicircles and lines in the frequency spectrum. Therefore, the information about dominant frequencies from $S_{rad}$ and dominant periodic patterns from $S_{ang}$ is used to generate features. From both sets $S_{rad}$ and $S_{ang}$, the mean, the maximum, the location of the maximum, the variance, and the difference between minimum and maximum values are extracted, yielding a feature vector with ten spectral features.

#### 2.4.2. Wavelet Features

The two-dimensional wavelet transformation is also explored in this work to perform frequency domain analysis. It is derived from the correlation between the image and the wavelet ψ. Integrating over $R^{2}$, the continuous two-dimensional wavelet transformation is defined as:(3)$$W_{f}\left(t,s,\theta \right)=\int f\left(p\right)\psi_{t,s,\theta }^{*}\left(p\right)dp$$
where t represents the translation vector, *s* the scaling parameter, and θ the rotation angle. The wavelet transformation is related to the Fourier transformation but also provides positional information by analyzing the frequency content within different resolutions, which is also called multiresolution analysis. It offers high positional accuracy for high frequencies, and offers high frequency resolution for low frequencies. As an example of a wavelet function, the simplest form, the Haar wavelet, is used. Three scales are used to produce 10 different channels, so that excitations can be detected by analyzing the image on different scales. As in the 2D FFT, for the 2D FWT the transformation needs to be executed on all rows first, and then on all columns. The processing on the rows and columns has to be performed subsequently before changing the level; otherwise, a different type of a wavelet transform will result.

When the 2D Haar wavelet transformation is applied, the image is decimated into four channels. The decimation of the image is illustrated in the bottom branch in Figure 4. In the output of a one-level Haar wavelet transform, the top left channel (Ea) is a smaller version of the original image, containing the low-frequency information, which is obtained by low-passing the original image, realized by a scaling function averaging neighboring values. The three remaining channels contain the high-frequency portions (or the wavelet portions) of the signal, which are the horizontal (*Hx*), vertical (*Vx*), and diagonal (*Dx*) edge information. For transforms with levels higher than one, the top left image is decimated recursively as described above.

To extract the wavelet features, the the energy of each channel is computed as follows:(4)$$E_{f_{d}}=\frac{1}{NM}\sum_{p_{x}=0}^{N-1}\sum_{p_{y}=0}^{M-1}\left|W_{f_{d}}\left(p\right)\right|,$$
with N−1 and M−1 as the image dimensions and $W_{f_{d}}\left(p\right)$ as the discrete Haar wavelet transform in position $p={\left(p_{x},p_{y}\right)}^{T}$ in the image. This results in a vector consisting of ten extracted wavelet features. Large energies in both low and middle frequencies can be observed in images with blob-like excitations, while the low-frequency channels are dominant in smoother images.

#### 2.4.3. Feature Analysis

Another way to optimize the feature extraction is to reduce the number of extracted features. In this section, we evaluate whether some features can be left out causing only a negligible loss in classification performance. For this, feature analysis using principal component analysis (PCA) is performed. The goal is to find out whether selecting only a subset of the generated features would reduce the classification performance significantly. With fewer features to extract, the complexity of the approach and the execution time could potentially be reduced significantly.

It is computationally expensive to evaluate all combinations of features. With *n* features, there would be $2^{n}-1$ combinations. An alternative is to select a subset of the features from the original set sequentially. The selection is performed by calculating the variance each feature accounts for in a principal component analysis. In practice, this means feature that accounts for the most variance is selected first, and then the other features are added incrementally. At each step, the accuracy is evaluated.

The results of these methods are illustrated in Figure 5. It shows that the accuracy of the spectral features is higher than the wavelet features. For the spectral features, the accuracy does not increase by a large margin after selecting four features. For the wavelet features, after selecting only six features, a saturation of the classification performance can be observed. This result can be explored further when optimizing the system performance. It could be the case that extracting fewer features reduces the execution time by a big factor.

### 2.5. Dataset, Training, and Classification

For the PAMONO sensor data, different types of datasets are available: there are datasets from signals of virus-like particles (VLPs) and polystyrene particles (PPs), for particle sizes of 200 nm and 100 nm, of which the 100 nm ones are harder to detect since they are both smaller and more faint. VLPs have the same structure and dimensions as real virus particles, and they also have the same binding behavior to antibodies [18]. The benefit of VLPs over real virus particles is the absence of viral genetic material inside VLPs. This fact makes the replication of VLPs impossible and particles non-infectious. Lacking the potential to cause infectious diseases, VLPs may be easily handled in any laboratory. On the other hand, PPs do not need antibodies for binding, and are thus easier to handle. PPs are also commercially available in different sizes. Only a small electrical charge is sufficient to bind them to the gold surface. The signal produced by PPs is slightly stronger than the signal of real viruses or VLPs of the same size [19], i.e., the refractive index (RI) of VLPs or viruses is 1.36–1.40 and it is closer to the RI of the sample solvent (water, RI = 1.33) than the RI of PPs, which is 1.54–1.60, depending on type of PP. In practice, this means that PP bindings are slightly easier to detect. However, PPs still induce representative signals that are useful for our analysis. For this reason, in this study, only datasets containing bindings from PPs were used for training and testing. The datasets were originally created using the Synthesis-Optimization-AnalySis (SynOpSis) approach proposed by Siedhoff in [1]. Since manually annotating the bindings is expensive, data from already annotated PP bindings were used to produce plausible synthetic datasets. For the experiments in this study, only synthetic data were used for training and testing.

The datasets used for training and testing the classifier were provided by Lenssen et al., and more detailed information on the sets can be found in [3]. The datasets were acquired by running the PAMONO image analysis pipeline up to the detection stage on full raw sensor images, such as the public images available on the SFB website [13], to produce image patches of size 48×48. The image patches were then annotated by experts with the labels “virus” or “no virus”. The total number of image patches is 38,871, and the number of those containing particles of sizes 100 nm and 200 nm is 38,871. The train–test split used was 19,526 images for training and 19,345 for testing.

For training the classifier on the extracted features, the decision tree (DT) and random forest (RF) libraries in sk-learn [20] were used. DTs with small depths are very fast classifiers, and it is convenient to obtain their C++ code for the use in embedded systems. When the classifier model is trained and is ready for operation, the tree structure and the split values of the tree from the trained model are extracted, and then converted to deployable C++ code, as proposed by Buschjäger et al. [21] as a standard if-else tree. The tree then is appended as the next step in the pipeline after feature extraction. The models can be executed after feature extraction with the spectral, wavelet, or both feature sets as input. At the end, a binary decision is returned as a result for every image patch processed in the image analysis pipeline. Note that the depths of all DTs are limited, and have a maximum depth no greater than 12. The execution time for the DTs with depths less or equal to 12 is negligible, since it is smaller than 1μs. In our experiments, increasing the depth of the DTs further does not increase classification performance by a large margin.

Please note that the DTs and RFs need to be trained anew for every platform to achieve the best accuracy possible. For example, to train a DT to classify on a certain model of an ARM CPU, the features for the training of the DT have to be extracted on this exact same ARM CPU. If a certain model of an Intel CPU is used, the ARM DT may not be compatible with the features extracted on the Intel CPU. For this exact same Intel CPU, the DT has to be trained anew.

### 2.6. Evaluation Metrics

The metrics precision, recall, and accuracy were used to evaluate the classification performance of the methods. To calculate the metrics, the numbers of *TP*, *FP*, *FN*, and *TN* are needed. *TP* stands for true positive, *FP* for false positive, *FN* for false negative, and *TN* for true negative. Precision is calculated as $\frac{TP}{TP+FP}$, and measures the ratio of correctly as positive detected elements, considering only positive predictions. *FP* in the denominator holds the information about the precision; if it is high, the precision will be low. However, most elements could be predicted to be false, and only a few correctly as positive, which would give a high precision value despite having many *FNs*. For this reason, a measure for the number of *FNs* is needed, which is recall. Recall is calculated as $\frac{TP}{TP+FN}$, and measures the ratio of correctly as positive detected elements, considering only elements that have positive ground truth annotations. If the number of *FNs* is high, the recall will be low. Accuracy is calculated as $\frac{TP+TN}{TP+FP+FN+TN}$, and measures the ratio between the correct predictions divided by all predictions. Since the dataset is symmetric, the accuracy can be used as a reliable summary metric for the classification performance.

## 3. Results

To evaluate the effectiveness of our classification, we measured the classification performance and execution time of our methods. To evaluate classification performance, we used precision, recall, and accuracy. We then compared our accuracy results with the CNNs by Lenssen et al. [3].

To evaluate execution time, we considered two experimental settings. In the first setting, for a fair comparison between the frequency analysis and the CNNs, we measured the execution time for computing the deepRacin graphs. The execution time for this setting was measured on an Intel Core i7-4600U with the integrated Intel HD Graphics 4400, to which the computations of the 2D FFT and 2D FWT were offloaded with OpenCL kernels. We accelerated the 2D FFT with shift and argument, and the 2D FWT with the energy computation.

We notice that offloading computation to GPUs requires synchronization with significant overhead. Hence, for the second setting, we did not accelerate any computations, and we evaluated a CPU-only implementation of the feature extractions. As a general purpose evaluation platform, we used the Intel Core i7-4600U without GPU acceleration. For testing on an embedded platform, we ran the feature extraction on the ARM Cortex-A9 chip on the Zedboard, again without any acceleration. We separately evaluated the xfft IP as an accelerator against the Fastest Fourier Transform in the West (FFTW) library on the Zedboard.

### 3.1. Classification Performance

As shown in Table 1, the classification shows the best results when both FFT and FWT features are used. When only one set of features is used, the FFT features outperform the FWT features by a small margin. Increasing the number of trees in a random forest improves the performance only by a small margin. When no restrictions are imposed on the tree depths, the classification performance increases around 0.15 percentage points for all cases. However, the tree depths used by the training algorithm in that case can amount to over 39, which can potentially use a lot of resources. A few results of classifications are shown in Figure 6. When the excitations are not strong enough, the classification might miss them, which counts as an *FN* error. When vibrations or noise disturb the signals, it can be mistaken as a particle binding (*FP*). *FP* errors occur when artifacts, noise, or vibrations cause regular high frequency patterns, and the method classifies them as a nanoparticle binding. Overall, the precision and recall of all our approaches are above 96%, and the accuracy is above 97%, except for FWT with one tree. The CNN [3] has a higher accuracy score (99.5%) than both feature approaches in Table 2.

### 3.2. Execution Time Using a GPU for Acceleration

In Table 2, the experiments show that our spectral analysis approach is 2.63 times faster than the CNN solution. The wavelet approach is 1.83 times faster than the CNN. We notice that the synchronization overhead between CPU and GPU dominates the execution time. The computation steps of the FFT classification of size 32×32 using the setting in Table 2 only take 10% of the whole execution time; the rest of it is needed for synchronization. The FWT has similar synchronization overhead.

To explore this issue further, we evaluated parts of the deepRacin graph for feature generation on an Intel Core i5 4460 with an AMD Radeon R7 370 as the accelerator. For the FFT of a 32×32 image on this platform, only 5% of the time is spent on computation, the rest is synchronization. We get similar ratios for the FWT.

### 3.3. Execution Time Results for CPU-Only Version

The implementation of the classification methods in deepRacin allowed us to compare the CNN and the feature-based methods in a fair setting, since the hardware and tools used were the same. However, we notice that the synchronization overhead is the main bottleneck in our feature-based method. Hence, we executed the classification as a CPU-only version on an Intel Core i7 4600U, which is the same CPU used to acquire the results in Table 2, without using the GPU for acceleration. To evaluate the execution time of the frequency domain analysis on an embedded platform, we executed the classification as a CPU-only version on the ARM Cortex-A9 as well. For the CPU-only version, we used the FFTW library for FFT, and a CPU version of the 2D Haar FWT. Our results show that the CPU-only executions give a significant speedup.

In Table 3, the decomposition of the runtime of our classifications are presented for different platforms. Only the time for executing the classification is considered, not the execution time needed to generate and load the test images. We executed the FFT and FWT algorithms (Alg.) on the Intel and ARM platforms (Platf.). After the transforms (Transf.), for the FFT, the post-transformation (PT) step calculates the arguments and computing the shift. The summing (Sum) part consists of $S_{rad}$ in the first value, $S_{ang}$ in the second value, and final calculations to obtain the 10 features in the third value. We notice that the computation of $S_{ang}$ accounts for around 70% of the execution time; the issue is observed in the measurements for ARM as well. For FWT, the main bottleneck is the 2D FWT. After the wavelet transform, the energies and their sums are computed; we see that the influence on the total execution time of the PT, Sum, and DT stages of the FWT classification is weak. Overall, the FWT classification outperforms the FFT classification in execution time in this CPU-only setting; the FWT method is 1.7 times faster than the FFT method.

We also observe that the execution time measured on the ARM platform is around 13 times higher than on the Intel platform for the FFT classification, and around 8 times higher for the FWT features. This is an expected result, as the ARM CPU is an embedded platform.

According to our feature analysis in Section 2.4.3, a trade-off may be possible between the maximal accuracy and the computation time with the aforementioned procedure. In this subsection, we further explore this trade-off with the results in Table 3. On the x-axis is the computation time for a subset of *f* features required to extract those *f* features. The features are ordered according to the aforementioned correlation in Section 2.4.3. As shown in Figure 7, compared to execution time needed to extract all 20 features, we can save around 83.5% of the computation time and achieve more than 93% accuracy by extracting only two FFT features (by leaving out the extraction of $S_{ang}$ completely). If we extract four more FFT features, we can save about 40% of the computation time and sacrifice only about 2% accuracy. The maximum classification accuracy using all 20 features is 97.78%, but the computation time is significantly increased from 60% to 100% in this case, because the FWT features have to be extracted as well.

### 3.4. FFTW against xfft IP

In the Xilinx SDSoC environment, the xfft IP core can be used, which executes a 1D FFT of any point size. We compared the use of the 1D xfft IP against the FFTW library on the Zedboard.

The 32-point xfft IP is called 64 times to perform a 2D FFT on the image patches. Calling only the xfft IP in this way needs 271,138 ns averaged over 1000 runs, while the FFTW, as a CPU-only version, needs 61,608 ns. The xfft IP execution time is limited by the data transfer overhead between the ARM chip and the PL, as well as the PL clock on the Zedboard, which is limited to 100 MHz. We therefore do not use the xfft IP for acceleration in our application. However, if the image size (and thus the FFT point size) were larger, or another, faster FPGA platform were used, there could be more benefit using the xfft IP, and the two methods should be compared anew.

To speed up data transfer, it is possible to extend the xfft IP core with a memory controller, which stores the entire patch and intermediate FFT results, with a design in a hardware description language (HDL), to perform a 2D FFT. However, how to optimize the execution in HDL is beyond the scope of this work. Here, we only consider readily available components.

## 4. Discussion

### 4.1. Experiment Results

In this study, we explored resource-efficient algorithms and execution platforms to run the virus classification of PAMONO biosensor images. The results show that using frequency domain features with a small decision tree classifier offers sufficient classification performance for virus classification. Compared to the CNN, the accuracy of our methods can be 1–2.5 percentage points lower. However, the execution time of the frequency domain approach is between 17 and 46 μs on a general purpose mobile platform. For comparison, the CNN needs 3370 μs. To explore the execution platform, we ran the feature extraction on a general purpose mobile Intel CPU + GPU platform, on a general purpose mobile Intel CPU-only platform, and on an embedded ARM platform, with CPU-only and PL accelerated configurations. The results show that a platform with a simple ARM Cortex-A9 CPU is sufficient to run the classification, without the need for additional acceleration hardware. The ARM version needs 88 μs for the FWT version, and 376 μs for the FFT version of the classification.

We conclude that the classification of virus particles can be run on platforms with strict resource requirements. We also conclude that the trade-off between execution time and classification performance can be used with the frequency domain-based classification, for either faster execution with less accuracy or slower execution with more accuracy. The execution time of our frequency domain methods can be also be measured on other platforms by using our published tool in [22].

### 4.2. On Using Different Platforms for Classification

The accuracy of the frequency analysis methods was confirmed in two ways: Lenssen et al. [3], obtained results in a MATLAB evaluation, and in this work we obtained similar accuracy values by implementing the feature extraction in deepRacin.

Different platforms, e.g., CPU and GPU, or ARM and Intel, produce different numerical errors in floating point arithmetic. Different platforms therefore may produce differing results for such operations due to floating point approximation errors. The FFT algorithm predominantly uses floating point operations, and in the FFT feature extraction we sum thousands of floating point values. In fact, we perform over 4500 floating point summing operations to obtain arrays $S_{rad}$ and $S_{ang}$. Additional sums are performed in the final feature extraction step, i.e., calculations of the mean and variance.

In these summations, the errors add up differently for different platforms, and the DT has to be adapted to each platform. A direct consequence is that the DTs trained on the features extracted in MATLAB cannot be used to classify the features extracted in deepRacin. MATLAB uses the CPU to calculate the transforms, while deepRacin uses the GPU. A new DT has to be trained by extracting features in deepRacin.

For the case of Intel and ARM (both CPU-only), as well as ARM with the xfft IP, to get the best accuracy results, we would need to train a DT by generating the features for training the DTs on every platform separately. Since the accuracy of the classification methods is confirmed in two different works, we only consider the execution time of our classification for the CPU-only evaluation. Furthermore, to preserve the portability of our evaluation framework for different platforms, we generate 1000 random grayscale images during the execution time test. During testing, we generate a batch of 25 random grayscale images at once, and extract the features. This procedure is repeated 40 times to reach 1000 runs. The source code of the testing framework is uploaded in our GitHub repository in [22]; it can be compiled and evaluated for several different platforms.

Although there is no difference in the number of executed instructions when using random images, the extracted features are not representative for the features that are extracted from the original patches. Therefore, we load 1000 extracted feature vectors into memory for FFT and FWT features, respectively, which were generated using deepRacin. Then, we use the trained DTs from the deepRacin implementation, to get accurate and representative features and execution times for the DT. This test flow allows us to evaluate the classification even on platforms with memory limitations, without having to extract the features of the training examples for every platform.

On a related note, zeropadding the input image causes interpolation values in the FFT, and introduces more errors in the summation part of the FFT classification. We encountered this problem when we zeropadded the 48×48 image patches to 64×64, to benefit from powers of two in the FFT and FWT algorithms. With such interpolation values, the errors add up more. The errors due to interpolation were directly noticeable in the accuracy of the FFT classification. The FFT classification trained on a dataset zeropadded to 64×64 showed around 10% less accuracy than the FFT classification trained on a dataset without zeropadding (32×32). On the other hand, the FWT classification showed no significant changes in accuracy when zeropadding to 64×64 (there is no interpolation for the FWT in this case; zero values stay zero values for FWT), or cropping to 32×32 from the 48×48 images.

## 5. Conclusions

In this work, we introduce new insights into the design possibilities of a mobile and low-power nanoparticle detection device. We explore resource-efficient algorithms and execution platforms to run the virus classification of PAMONO biosensor images. We show that the frequency analysis based nanoparticle classification can be realized using readily available, low cost components, without the need for additional hardware accelerators, such as GPUs or FPGAs. The execution time is improved by a large factor compared to the previous work, in the range of 17–29 μs per image. The classification accuracy of our frequency analysis based approaches is competitively high compared to the CNN method, which has an accuracy of approximately 1–2.5 percentage points higher, with 3370 μs execution time. Given the same amount of time, our proposed methods can classify around 120–200 times more image patches than the CNN.

Our work explores the design space of nanoparticle classification systems with frequency analysis methods and points the way toward a faster and more resource-efficient alternative to CNNs. In the future, we plan to evaluate efficient methods for the detection stage, improve our feature extraction, and include the classification of smaller particles, to build a small, fast, low-power and low-cost embedded system for nanoparticle classification.

## Figures and Tables

**Figure 1 sensors-19-04138-f001:**
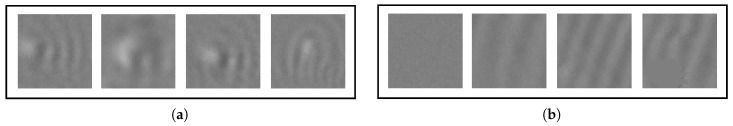
Images of positive samples in (**a**) and negative samples in (**b**).

**Figure 2 sensors-19-04138-f002:**
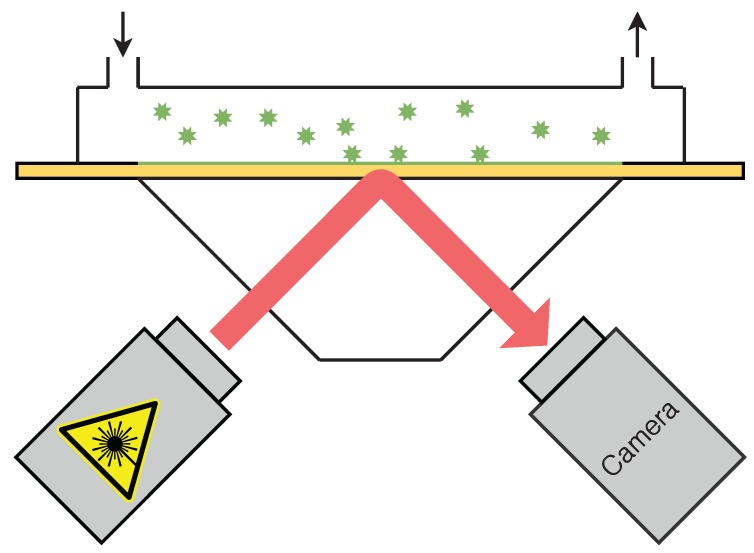
A sketch of the PAMONO biosensor setup.

**Figure 3 sensors-19-04138-f003:**
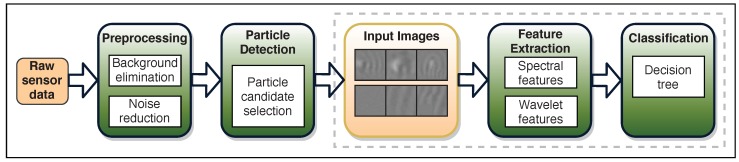
The image processing pipeline.

**Figure 4 sensors-19-04138-f004:**
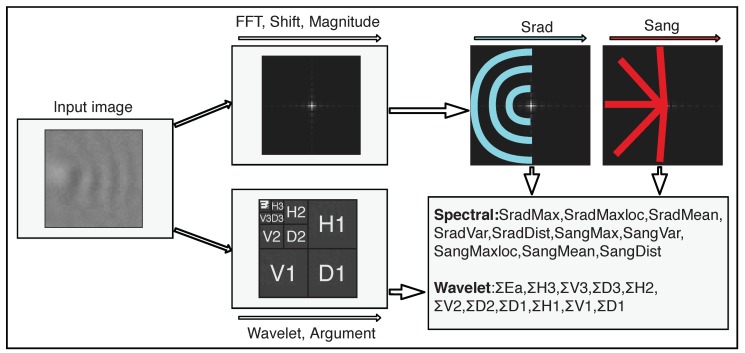
The main steps of feature extraction process. In the top branch, the extraction of Fourier features is illustrated. In the bottom branch, the extraction of Haar wavelet features is illustrated.

**Figure 5 sensors-19-04138-f005:**
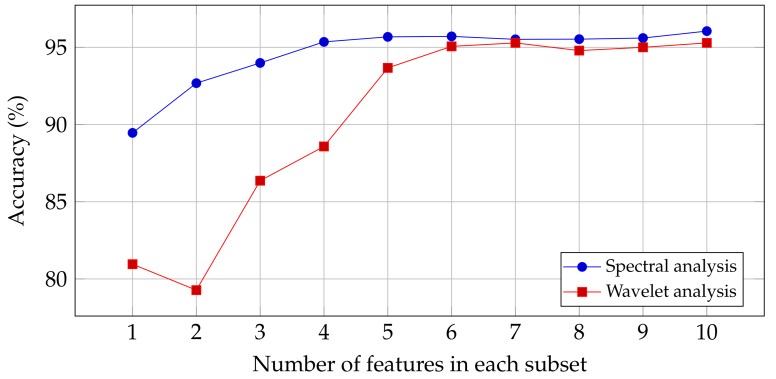
Accuracy of classification and average normalized computation time for different subsets of features.

**Figure 6 sensors-19-04138-f006:**
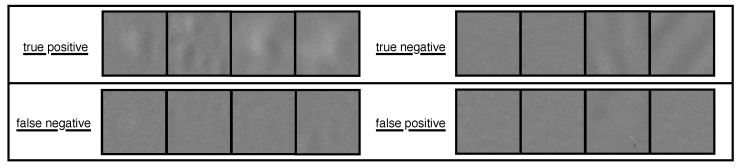
True positive (*TP*), true negative (*TN*), false negative (*FN*), and false positive (*FP*) for the approach with spectral and wavelet features.

**Figure 7 sensors-19-04138-f007:**
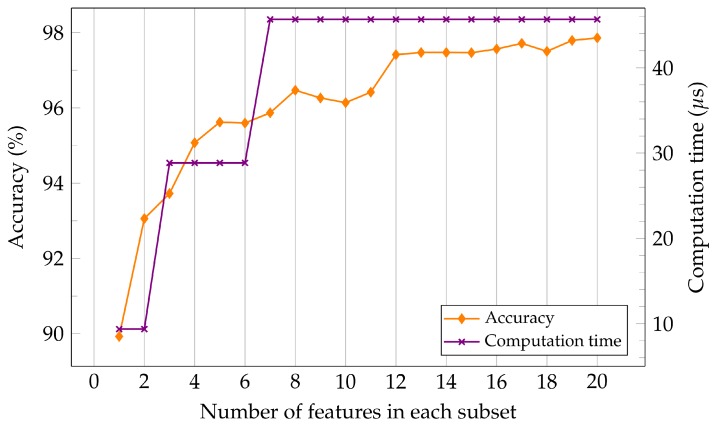
Trade-off between accuracy and computation time for different subsets of features.

**Table 1 sensors-19-04138-t001:** Comparisons between different classifiers using different feature compositions. DT, decision tree; RFn, random forest with n DTs; P, precision; R, recall. The results are based on the work in [9].

Features	Measure	DT (%)	RF10 (%)	RF100 (%)
Only FFT	P, R	97.78, 96.33	98.50, 96.02	98.66, 96.48
Only FWT	P, R	96.77, 96.35	97.59, 96.63	97.66, 97.49
FFT + FWT	P, R	98.49, 97.04	99.33, 97.19	99.33, 97.76

**Table 2 sensors-19-04138-t002:** Average execution time (Intel Core i7 4600U with integrated Intel HD Graphics 4400), and accuracy (with one DT for the feature based approaches). The results are based on the work in [9].

Method	Accuracy (%)	Execution Time (ms)
FFT features	97.07	1.28
FWT features	96.57	1.50
FFT and FWT features	97.78	2.78
CNN [3]	99.50	3.37

**Table 3 sensors-19-04138-t003:** Compositions of the execution time for different platforms. All values are average values obtained from 1000 runs, and are in  ns.

Alg.	Platf.	Transf.	PT	Sum	DT	Total
FFT	Intel	3563	3415	2367 + 19,438 + 19	43	28,848
	ARM	61,801	43,302	21,902 + 246,765 + 975	1310	376,058
FWT	Intel	16,594	160	21	45	16,821
	ARM	79,612	6271	934	1123	87,942

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
