# Peer review of "Nanoparticle Classification Using Frequency Domain Analysis on Resource-Limited Platformsâ€"

_sensors, 2019, doi:10.3390/s19194138_

Round 1
Reviewer 1 Report
This manuscript addresses an important technical issue in the detection of the existence of viruses and virus-like particles by using PAMONO. The authors demonstrate clearly that their proposed methods can classify around 120 to 200 times more image patches than the CNN.
In conclusion, this manuscript appears to be reporting some signal processing measurements made on the PAMONO sensor, however, the impact is lost because it has not been verified in PAMONO sensor experiments with biological samples. There are some comments on the PAMONO sensor experiment. Comment 1. What sample did you use in PAMONO sensor experiment?
2. Did you demonstrate the detection of actual virus in PAMONO sensor experiment? I think that you should indicate the advantage of your proposed method by using an actual virus or biological vesicles such as previously studied [1].
[1] Application of the PAMONO-Sensor for Quantification of Microvesicles and Determination of Nano-Particle Size Distribution Sensors 2017, 17(2), 244;
The manuscript would be improved by a thorough English language review before acceptance for publication.
・line40: allow to deploy → allow deploying
・line84: The core area of this work are → The core area of this work is
・line93: webite→ website
・line231: The results of this methods→ The results of these methods
・line298: classifictions are shown in Figure 6.→ classifications are shown in Figure 6.
・line386: The accuracy of the frequency analysis methods were shown→ accuracies or was
Reviewer 2 Report
This work proposed a resource-efficient algorithms and execution platforms for virus classification of PAMONO biosensor images. It could facilitate the portability of the biosensor. This algorithms greatly reduces execution time. However, its accuracy is a little lower than CNN. The accuracy of the sensor is more important for users. The authors should considerate the practical application of the sensor.
In generally, the text is written in a style that makes it hard to read. The presentation of this article needs to be revised.
